# Activation of DNA Damage Tolerance Pathways May Improve Immunotherapy of Mesothelioma

**DOI:** 10.3390/cancers13133211

**Published:** 2021-06-27

**Authors:** Hélène Brossel, Alexis Fontaine, Clotilde Hoyos, Majeed Jamakhani, Mégane Willems, Malik Hamaidia, Luc Willems

**Affiliations:** 1Molecular and Cellular Epigenetics, Interdisciplinary Cluster for Applied Genoproteomics (GIGA), University of Liège, 4000 Liege, Belgium; Helene.Brossel@uliege.be (H.B.); alexis.fontaine@doct.uliege.be (A.F.); Clotilde.Hoyos@uliege.be (C.H.); majeed.jamakhani@uliege.be (M.J.); megane.willems@uliege.be (M.W.); mhamaidia@uliege.be (M.H.); 2Molecular Biology, Teaching and Research Centre (TERRA), Gembloux Agro-Bio Tech (GxABT), University of Liège, 5030 Gembloux, Belgium

**Keywords:** mesothelioma, tolerance to DNA damage, DNA damage response, chemoresistance, immunotherapy, immune checkpoint inhibitor

## Abstract

**Simple Summary:**

A critical step in the success of immunotherapy is the presentation of tumor-derived peptides by the major histocompatibility complex I (MHC-I) of tumor cells. These neoantigens are potentially immunogenic and trigger immune responses orchestrated by cytotoxic cells. In malignant mesothelioma (MM), tumor development is nevertheless characterized by a low mutation rate despite major structural chromosomal rearrangements driving oncogenesis. In this paper, we propose a paradigm based on the mechanisms of the DNA damage tolerance (DDT) pathways to increase the frequency of non-synonymous mutations. The idea is to transiently activate the error-prone DDT in order to generate neoantigens while preserving a fully competent antitumor immune response.

**Abstract:**

Immunotherapy based on two checkpoint inhibitors (ICI), programmed cell death 1 (PD-1, Nivolumab) and cytotoxic T-lymphocyte 4 (CTLA-4, Ipilimumab), has provided a significant improvement in overall survival for malignant mesothelioma (MM). Despite this major breakthrough, the median overall survival of patients treated with the two ICIs only reached 18.1 months vs. 14 months in standard chemotherapy. With an objective response rate of 40%, only a subset of patients benefits from immunotherapy. A critical step in the success of immunotherapy is the presentation of tumor-derived peptides by the major histocompatibility complex I (MHC-I) of tumor cells. These neoantigens are potentially immunogenic and trigger immune responses orchestrated by cytotoxic cells. In MM, tumor development is nevertheless characterized by a low mutation rate despite major structural chromosomal rearrangements driving oncogenesis (*BAP1*, *NF2*, *CDKN2AB*). In this opinion, we propose to investigate an approach based on the mechanisms of the DNA damage tolerance (DDT) pathways to increase the frequency of non-synonymous mutations. The idea is to transiently activate the error-prone DDT in order to generate neoantigens while preserving a fully competent antitumor immune response.

## 1. Oncogenesis of Mesothelioma Occurs at Low Tumor Mutational Burden

Malignant mesothelioma (MM), a poor-prognosis cancer closely associated with asbestos exposure, affects mesothelial cells from the pleura, pericardium, and peritoneum [1]. The current model of MM oncogenesis postulates that neoplastic transformation is caused by a combination of genetic and epigenetic events leading to unresolved chronic inflammation. Asbestos fibers directly stimulate the production of reactive oxygen and nitrogen species (ROS and RNS) by mesothelial cells [2]. Iron associated with asbestos generates highly reactive hydroxyl radicals (HO·) from hydrogen peroxide (H_2_O_2_) through the Fenton reaction with oxidation of Fe^2+^ to Fe^3+^. In hypoxic conditions, asbestos also promotes RNS such as nitric oxide (·NO) and peroxynitrite (·ONOO) via the respiratory chain. ROS/RNS affect phosphodiester bonds of the DNA backbone, possibly leading to error-prone repair by non-homologous end joining (NHEJ) [3]. This mechanism may explain the presence of deletions and insertions in MM tumors (Figure 1A). Asbestos fibers also undergo “frustrated phagocytosis” by macrophages, further increasing the oxidative burden [4]. Besides cleavage of phosphodiester bonds, ROS directly oxidize DNA bases such as guanine into 8-oxoguanine (8-oxoG) [5]. If improperly repaired, the lesion results in mismatched pairing with adenine leading in G-to-T and C-to-A substitutions in the genome. In MM, however, the most frequently encountered mutation is C-to-T transition accounting for 49.56% of substitutions (Figure 1B,C) [6]. This type of mutation is thought to arise from deamination of 5-methylcytosine in areas of exposed single-stranded DNA. The mechanism, called kataegis, is mediated by the AID/APOBEC family of enzymes that deaminates cytosine to uracil [7,8]. Localized hypermutations in a small region of DNA have been identified in other cancer genomes. Although signatures of APOBEC-induced mutagenesis are not clearly identified in MM [9], it can nevertheless be speculated that the presence of single-stranded DNA associated with DNA replication rather than base oxidation is a major driver of oncogenesis in MM.

For still unclear reasons, the tumor mutational burden (TMB), which is defined as the total number of somatic mutations in the coding region of a genome, is particularly low in MM compared to other cancer types [9,10]. According to TCGA data, genomic analysis of MM tumors reveals that a median number of 0.726 somatic mutations occur per megabase (Figure 1D).

Hypermutation and genomic instability resulting from defective DNA mismatch repair (MMR) is infrequent in MM [11]. Instead, major genomic alterations affect a series of tumor-suppressor genes in human MM, e.g., the BRCA1-associated deubiquitylase (*BAP1*), the cyclin-dependent kinase inhibitor 2A/B (*CDKN2AB*), neurofibromatosis type 2/moesin-ezrin-radixin-like protein (*NF2*/*Merlin*), and tumor protein 53 (*TP53*). The combined deletion of *BAP1*, *CDKN2AB*, and *NF2* leads to a rapid onset of MM in mice [12]. *BAP1* is an enzyme that mediates the deubiquitination of histone H2A monoubiquitinated at lysine 119 (H2AK119ub1) [13]. *BAP1* also interacts with the BRCA1-BARD1 heterodimers and interferes with their ubiquitination activity. *BAP1* loss in mice results in increased expression of the enhancer of zeste-homolog 2 (*EZH2*), the trimethyltransferase of lysine 27 of histone H3 (H3K27me3). Consistently, MM cells that lack *BAP1* are sensitive to *EZH2* inhibition. Clinical evidence for efficacy of an *EZH2* inhibitor (Tazemetostat) was observed in a multicenter phase 2 trial on relapsed or refractory MM with *BAP1* inactivation [14]. Another frequently inactivated gene, *CDKN2AB*, encodes the ADP-ribosylation factor (ARF, p14), INK4A (p16), and INK4B (p15) via alternative open reading frames. By inhibiting cyclin-dependent kinases 4 and 6 (CDK4 and CDK6), INK4A affects the transition between the G1 and S phases of the cell cycle. The CDK4/CDK6 inhibitor Abemaciclib induces apoptosis in *CDKN2A*-mutated cells and suppressed tumor growth in a mouse model, leading to a clinical trial (NCT03654833) [15]. *NF2*/*merlin* is involved in contact inhibition by interacting with membrane-associated proteins such as CD44, α/β-catenin, and actin fibers [16]. A loss of merlin expression disrupts cancer-related signaling through the Hippo and mTOR pathways.

Major alterations in *BAP1*, *CDKN2AB*, and *NF2* are thus predicted to drive oncogenesis and provide opportunities for targeted therapies. Notwithstanding these recurrent genomic changes, genome-wide somatic mutations are thus relatively infrequent in MM.

## 2. High-Dose Treatment with Cisplatin and Pemetrexed Selects Chemoresistant Mesothelioma Cells

Standard-of-care chemotherapy for MM patients is based on the combination of a DNA crosslinking agent, cisplatin, and an antifolate, pemetrexed [17]. After aqua activation in the cytoplasm, cisplatin induces DNA adducts through covalent bonds and intrastrand crosslinks, which block the DNA replication machinery in the S phase of the cell cycle. Pemetrexed is a multifolate antagonist that impairs the *de novo* synthesis of tri-phosphate deoxyribonucleotides (dNTPs) through inhibition of thymidylate synthase (TS), dihydrofolate reductase (DHFR), and glycinamide ribonucleotide formyltransferase (GARFT), thereby inhibiting DNA synthesis, cell replication, and DNA repair [15,18,19]. The combination of cisplatin and pemetrexed also induces single-strand breaks (SSB) that are converted into double-strand breaks (DSBs) upon DNA replication.

Despite a relatively low efficacy, the combination of cisplatin and pemetrexed has remained the palliative therapy of MM for almost two decades. This regimen slightly extends the median overall survival of MM patients to 14 months but is associated with a lack of response in a significant proportion of patients as well as quick relapse.

Resulting from genetic mutations, transcriptional changes, or epigenetic modifications, resistance to cisplatin is multifactorial [18,19]. The mechanisms of resistance notably include reduced intracellular accumulation due to inhibition of uptake and/or increase in efflux, as well as intracellular inactivation by thiol-containing molecules (i.e., scavengers) and DNA damage repair (DDR) (Figure 2).

Cisplatin can be uptaken by cells through passive diffusion across the plasma membrane, under a chloride ion gradient. Cisplatin also penetrates cells via the copper transporter 1 (CTR1), whose expression inversely correlates with chemoresistance [20]. Following aqua activation in the cytoplasm, cisplatin may undergo inactivation by scavengers such as glutathione (GSH) and cysteine-rich metallothionein. The interaction between aquated cisplatin and GSH occurs non-enzymatically via a conjugation reaction or can be catalyzed by GSH-S-transferase (GSTp). Another cisplatin-resistance mechanism is associated with an increased capacity to correct the DNA lesions via the transcription-coupled nucleotide excision repair (TC-NER). In contrast, the MMR complex does not directly process cisplatin adducts. However, components of the MMR pathway (e.g., hMSH2 and hMutS) directly recognize intrastrand adducts of cisplatin and are therefore critical for the maintenance of genome integrity. Although infrequent in MM, a deficiency in the MMR system (i.e., hML1 and hMSH2) is nevertheless potentially associated with resistance to cisplatin [21]. A predominant mechanism of cisplatin resistance results from its efflux out of the cell via copper-exporting P-type ATPases 1 and 2 (ATP7A and ATP7B) [18]. Furthermore, overexpression of the ATP binding cassette (ABC) ATPase-like multidrug resistance-associated MRP2 transporter is also associated with cisplatin resistance by extracellular export of platinum-GSH conjugates via an ATP-dependent mechanism. To a lesser extent, the resistance to cisplatin may further involve a complex interplay of a variety of pathways (PI3K/Akt, HER2/neu, MAPK), p53 inactivation, and overexpression of anti-apoptotic proteins.

Because pemetrexed has been frequently combined with other drugs, the mechanisms of resistance have been less well characterized. These include impaired cell entry, defective polyglutamylation, intracellular inactivation, and overexpression of folate enzymes (i.e., TS, GARFT, DHFR), as well as enhanced efflux [22]. Pemetrexed is internalized by three transmembrane receptors: the reduced folate receptor (RFC), the folate receptor-α (FR-α), and the proton-coupled folate receptor (PCFT). In the cytoplasm, pemetrexed is activated by the folylpolyglutamate synthetase (FPGS). The glutamate tails of this active form are hydrolyzed by the γ-glutamyl hydrolase (GGH) in the lysosome. The de-polyglutamated pemetrexed is thereafter exported by members of ABC transporters (i.e., MRP).

Together, these different pathways select chemoresistant cells in MM patients treated with standard doses of cisplatin and pemetrexed.

## 3. Immune Checkpoint Blockade Therapy Requires Altered-Self Antigens

Considering the disappointing survival rate of MM patients following chemotherapy, other strategies have been investigated in recent years. Immunotherapy based on checkpoint inhibitors (ICI) has provided a significant improvement in overall survival for previously untreated unresectable MM [23]. In the recent CheckMate 743 trial, the combination of anti-programmed cell death 1 (PD-1, CD279) and anti-cytotoxic T-lymphocyte 4 (CTLA-4, CD152) antibodies (Nivolumab and Ipilimumab, respectively) increased the 2-year overall survival rate by 50% compared to platinum plus pemetrexed chemotherapy [24]. Despite this major breakthrough, the median overall survival of patients treated with the two ICIs only reached 18.1 months (vs. 14 months in standard chemotherapy). With an objective response rate of 40%, only a subset of patients benefited from Nivolumab and Ipilimumab immunotherapy.

This partial response may be explained by the low mutational burden encountered in MM. According to data available in other cancers, such as melanoma or non-small cell lung cancer (NSCLC), responders to ICI therapy typically present high numbers of somatic mutations in genomic coding regions [25,26]. The underlying prediction is that mutated genes generate altered-self proteins that can be processed in the form of small peptides, referred to as neoantigens, and presented on the major histocompatibility complex I (MHC-I) molecules at the surface of tumor cells. Only a small fraction of the non-synonymous mutations will produce new peptides loaded and presented by MHC-I (Figure 3B). These mutated peptides are potentially immunogenic and trigger immune responses orchestrated by CD8^+^ T cells, natural killer cells (NK), and macrophages. It is therefore possible that the generation of neoantigens may increase the immune infiltration within the tumor microenvironment.

The cellular composition of the MM tumor microenvironment is quite heterogeneous and consists of endothelial, stromal, and immune cells such as tumor-associated macrophages (TAMs), tumor-infiltrating lymphocytes (TILs), myeloid-derived suppressor cells (MDSCs), granulocytes, and NK cells [27]. While T-cells, B-cells, and NK cells are generally associated with an antitumor response, TAMs and MDSCs mainly exert protumoral functions [28]. TAMs inhibit immune responses notably through the expression of immunosuppressive cytokines (e.g., TGF-β, IL-10). Their accumulation is associated with a poor prognosis in MM, as observed in other cancers. TAMs can nevertheless exert antitumor functions. Besides their ability to phagocytize and process foreign antigens, macrophages are indeed directly cytotoxic to MM cells [29]. Besides a broad spectrum in the tumor cell composition, MM is also characterized by a spatial heterogeneity, displaying a continuum between “hot” and “cold” profiles defined by high and low lymphocyte infiltrations, respectively (Figure 3A).

Among TILs, cytotoxic CD8+ T-cells represent the predominant population in MM and are key mediators of the antitumor response [30]. However, T-cells in tumors become exhausted under chronic stimulation of the T-cell receptor (TCR) combined with co-inhibitory signaling [31]. Of note, this dysfunctional state is molecularly distinct from T-cell anergy, which corresponds to a state of non-responsiveness following an antigen encounter. Instead, exhaustion is characterized by a progressive and hierarchical loss of effector functions including cytokine production, proliferative capacity, and cytotoxic activity. A major feature of this state includes the upregulated expression of multiple co-inhibitory receptors such as CTLA-4, PD-1, lymphocyte-activation gene 3 (LAG-3, CD223), and the T-cell immunoglobulin and mucin domain containing-3 (TIM-3, CD366) [32]. CTLA-4 is transiently upregulated on naive T-cells upon priming by antigen-presenting cells (APCs). CTLA-4 interacts with the co-stimulatory receptors CD80 (B7-1) and CD86 (B7-2) expressed by macrophages, dendritic cells (DCs), and B-cells, preventing their binding to CD28 and subsequent T-cell activation and proliferation. PD-1 attenuates the signaling from the TCR, which is activated by its interaction with the peptide-MHC-I complex [33]. Engagement of PD-1 at the surface of T-cells with PD-L1/PD-L2 expressed by tumor cells indeed results in immune suppression. PD-1 is also expressed by B-cells, NK cells, DCs, monocytes, and macrophages. Antibodies targeting PD-1 or CTLA-4 disrupt intercellular interactions and prevent exhaustion of CD8+ T cells [34,35]. The challenge of the ICI immunotherapy is therefore to temper cytotoxic T-cell exhaustion [33,36]. However, a prerequisite of immune checkpoint therapy is the presence of immunogenic-mutated peptides eliciting T-cell responses.

## 4. Mutations Generated by the DNA Damage Tolerance Pathways May Promote Neo-Antigen Production by MM Cells

Compared to other cancers, MM tumors are characterized by a low mutation rate despite major structural chromosomal rearrangements driving oncogenesis. On the other hand, lesions such as cisplatin adducts induced by high-dose standard chemotherapy initiate the DNA damage response [37]. The mode of DNA repair depends on the lesion: cisplatin-induced intrastrand crosslinks are primarily repaired by nucleotide excision repair (NER), while 8-oxoG are processed by 8-oxoguanine glycosylase and base excision repair (BER) [3,20]. Inadequate base pairing is also repaired by the MMR pathway [38]. Finally, DNA double-strand breaks induced directly by oxidation or resulting from cisplatin adducts are processed by error-prone NHEJ, error-free homologous recombination (HR), or single-strand annealing (SSA) depending on the phase of the cell cycle [39].

This complex interplay of repair systems maintains genomic integrity and promotes survival of MM tumor cells. If unrepaired by the DDR, the lesions can be further processed by the DNA damage tolerance (DDT) pathways (Figure 4).

DDT provides escape pathways to restart the stalled fork, resulting from unrepaired cisplatin adducts, and to resume DNA replication [40,41]. Thus, the damage is tolerated, allowing further cell proliferation despite accumulation of DNA lesions. Nowadays, these pathways are well characterized in *Saccharomyces cerevisiae* but remain rather unclear in humans. These mechanisms depend on the type of ubiquitination of the proliferating cell nuclear antigen (PCNA) [41,42]. The PCNA processivity factor undergoes mono-ubiquitination on Lys-164 by a complex of RAD6 (an E2-ubiquitin conjugase) and RAD18 (an E3-ubiquitin ligase) in response to replication fork stalling. The Lys-164 mono-ubiquitination of PCNA, which is tightly controlled by BRCA1 and BAP1, activates the error-prone translesion synthesis (TLS) pathway [43]. Then, the low-fidelity polymerase Pol η first incorporates a nucleotide in front of the damage [44]. Due to its high error rate (3.5 × 10^−2^), this polymerase misincorporates dGTP opposite dT [45]. Of interest, Pol η expression is upregulated in presence of cisplatin treatment, thereby promoting TLS. Another TLS polymerase, REV1, specifically incorporates dCTP opposite dG and abasic sites. Then, Pol ζ (REV3) extends the newly formed strand beyond the damaged site despite the spatial distortion associated with the lesion. Another set of enzymes, UBC13/MMS2 (E2-ubiquitin conjugases) in complex with HLTF or SHPRH (orthologs of Rad5 E3-ubiquitin ligase), extends the ubiquitin attached to PCNA into a Lys-63-linked poly-ubiquitin chain [46]. This post-translational modification activates two template-switching pathways: homologous recombination (HR) or fork reversal (FR). Using the intact sister chromatid as a template to allow for the DNA replication of the damaged strand, the template-switching pathways are error free. In the HR pathway, which is controlled by BRCA1/2, the recombinase RAD51 initiates strand invasion, while RAD52 promotes the annealing of the complementary strands. RAD54 forms a D-loop that leads to the formation of Holliday junctions, which can be resolved by the helicase BLM and the topoisomerase TOP3. In the FR pathway, the fork-remodeling helicases ZRANB3, SMARCAL1, and HLTF are recruited to reverse the stalled fork. RAD51 binds to the blocked strand and leads to the annealing of the two newly synthesized strands to create a “chicken foot” structure, while BRCA2 protects the fork from nucleases. Finally, DNA replication can resume through HR or branch migration [47].

DDT thus provides mechanisms to tolerate DNA lesions during replication, thereby increasing cell survival despite mutational burden.

## 5. Fine-Tuning of the DNA Damage Tolerance Pathways May Improve Immunotherapy

Despite major genomic alterations in cancer driver genes (*BAP1*, *NF2*, *CDKN2AB*), mesothelioma is unfortunately characterized by low mutation rates (Figure 1). Furthermore, only a small fraction of the non-synonymous mutations will generate new peptides presented by the MHC-I (Figure 3). Host immunity against these potentially immunogenic neoantigens is also tempered by the immunosuppressive environment of the tumor. Standard-of-care platinum-based chemotherapy is predicted to promote mutagenesis but initiates mechanisms of resistance acquired through high-dose drug exposure (Figure 2). Among a broad diversity of mechanisms, the DDR pathways are able to efficiently repair the DNA lesions in MM tumors. If unrepaired by the DDR, the lesions can be further processed by the DDT pathways (Figure 4).

In this context, we propose an approach based on the stimulation of error-prone DDT pathways using suboptimal doses and/or fine-tuned delivery of chemotherapy (Figure 5). Administration of the DNA-damaging drugs would transiently activate DDT mechanisms using error-prone TLS polymerases that are intrinsically mutagenic. Among cancer cytotoxic compounds, cisplatin is more mutagenic than carboplatin and oxaliplatin [48,49]. Although scarce information is available in the literature, cisplatin and oxaliplatin have been shown to induce TLS in non-small cell lung cancer and in gastric adenocarcinoma [50,51]. Besides cisplatin, low doses of cyclophosphamide, doxorubicin, or gemcitabine could also be used to concomitantly elicit immunogenic cell death (ICD). The problem is that, at clinically relevant doses, these chemotherapeutic drugs are toxic to T cells, NK cells, and DCs, thereby limiting their association with ICI. The drug dose and the delivery schedule can be optimized to trigger ICD and concomitantly preserve a sustained antitumor immune response [52]. The idea is to increase the frequency of non-synonymous mutations that will generate new immunogenic peptides. Host immunity would only be marginally affected by low doses or fine-tuned delivery of DNA-damaging drugs. Presentation of tumor-specific neoantigens by the MHC-I would efficiently stimulate novel immune interactions. Appearance of these neoantigens would occur in the absence of significant toxicity, notably for immune cells. We believe that transient activation of the DDT in tumor cells would increase the efficacy of immune checkpoint blockade and promote durable tumor regression.

## 6. Conclusions

Considering the spectacular promises of immunotherapy in MM, the major challenge will be to increase the proportion of responders and to prolong their survival. To this end, a number of approaches have been proposed [36]: (i) target additional immunomodulatory molecules (VISTA, LAG-3, TIM-3) [28]; (ii) facilitate infiltration of effector T-cells in tumors by using cytokines (e.g., IL-2) and growth factors (e.g., GM- CSF); (iii) inhibit regulatory T-cells (Tregs); (iv) induce immunogenic cell death (ICD) with chemotherapy (e.g., gemcitabine, doxorubicin) [52,53]; (v) combine ICI with radiotherapy (abscopal effect) [54,55]; (vi) use oncolytic viruses to lyse tumors [56]; (vii) improve antigen presentation with epigenetic modifiers (e.g., HDAC inhibitors) [57]; and (viii) target DNA-repair proteins (e.g., PARP or mth-1) [58,59].

In this opinion paper, we propose an alternative strategy considering that the central mechanism that initiates the antitumor immune response is the presentation of neoantigens by the TCR. The idea is to promote occurrence of somatic mutations by transient activation of error-prone DDT pathways and increase the frequency of antitumor peptides presented by the MHC-I. In this paradigm, it is essential to preserve optimal host immunity by reducing the doses of DNA-damaging agents and fine-tuning the kinetics of drug delivery. We are aware that a number of issues remain to be solved. A potential threat is the insufficient response to low or fine-tuned drug delivery followed by chemoresistance and tumor escape. Another issue is the complex interplay between overlapping DDR and DDT pathways that would limit the onset of novel somatic mutations. In this context, interference with MMS2/Ubc13/HLTF may favor TLS by reducing template switching [60].

## Figures and Tables

**Figure 1 cancers-13-03211-f001:**
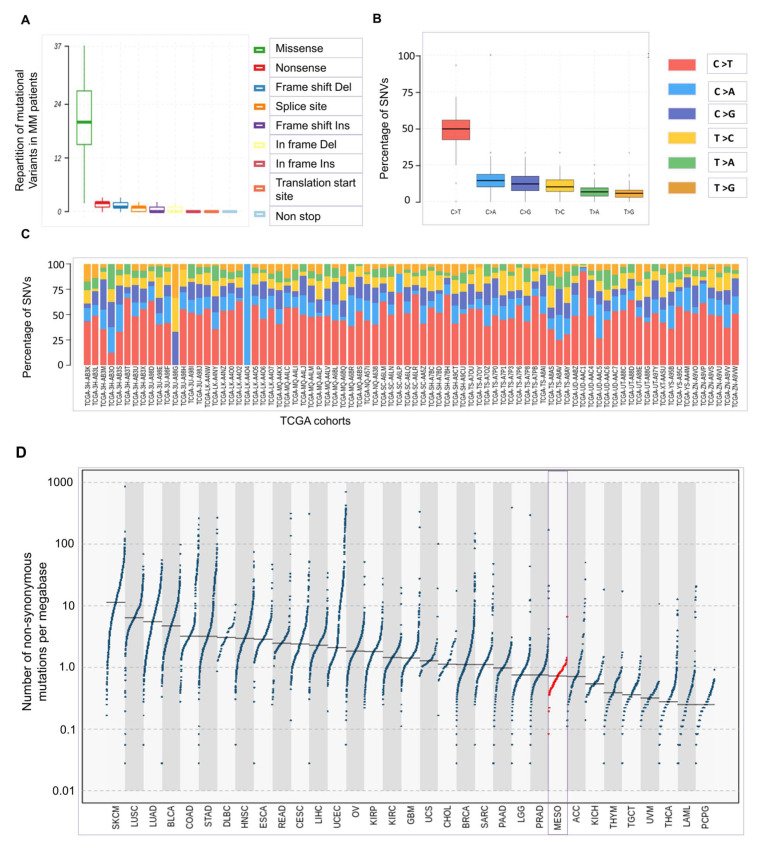
Low tumor mutational burden in mesothelioma**.** (**A**) Repartition of mutational variants in malignant mesothelioma (MM) patients. Data are expressed as median +/− interquartile. Abbreviations: deletion (Del) and insertion (In). (**B**,**C**) Percentage of single nucleotide variant (SNV) types among MM patients. (**D**) The number of genomic mutations per megabase was analyzed in different cancer types. The *Y*-axis represents the number of genomic nonsynonymous mutations per megabase of exons. Dots correspond to tumor samples and bars show median values. The different cancer types are indicated on the *X*-axis: skin cutaneous melanoma (SKCM), lung squamous cell carcinoma (LUSC), lung adenocarcinoma (LUAD), bladder urothelial carcinoma (BLCA), colon adenocarcinoma (COAD), lymphoid neoplasm diffuse large B-cell lymphoma (DLBC), stomach adenocarcinoma (STAD), esophageal carcinoma (ESCA), head and neck squamous cell carcinoma (HNSC), rectum adenocarcinoma (READ), cervical squamous cell carcinoma and end cervical adenocarcinoma (CESC), liver hepatocellular carcinoma (LIHC), uterine corpus endometrial carcinoma (UCEC), ovarian serous cystadenocarcinoma (OV), kidney renal papillary cell carcinoma (KIRP), glioblastoma multiforme (GBM), kidney renal clear cell carcinoma (KIRC), uterine carcinosarcoma (UCS), sarcoma (SARC), breast invasive carcinoma (BRCA), cholangiocarcinoma (CHOL), pancreatic adenocarcinoma (PAAD), brain lower-grade glioma (LGG), adrenocortical carcinoma (ACC), prostate adenocarcinoma (PRAD), mesothelioma (MESO, in red), kidney chromophobe (KICH), testicular germ cell tumors (TGCT), thymoma (THYM), acute myeloid leukemia (LAML), uveal melanoma (UVM), thyroid carcinoma (THCA) and pheochromocytoma and paraganglioma (TCPG). Data were extracted using TCGAbiolinks version 2.1 from the TCGA-GDC repository and plotted using maftools version 2.7.40 and RStudio 4.0. Bars represent median values.

**Figure 2 cancers-13-03211-f002:**
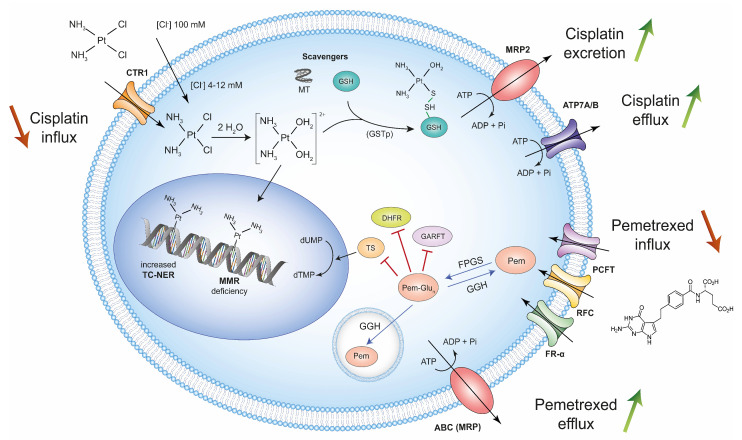
Main mechanisms of chemoresistance to cisplatin and pemetrexed. Reduced expression of copper transporter 1 (CTR1) leads to a decrease in the cisplatin influx. Aquated cisplatin in the cytoplasm can either generate intrastrand adducts with DNA or be inactivated by metallothionein (MT) or be conjugated with glutathione (GSH) by GSH-S-transferase p (GSTp). In response to DNA-cisplatin adducts, increased transcription-coupled nucleotide excision repair (TC-NER) activity and mismatch repair (MMR) deficiency can lead to cisplatin resistance. Upon inactivation, GSH-cisplatin conjugates will be excreted by the ATP binding cassette (ABC) ATPase-like multidrug resistance-associated (MRP2) transporter exported by the copper-exporting P-type ATPases 1 and 2 (ATP7A/B). Pemetrexed influx is regulated by the proton-coupled folate receptor (PCFT), the reduced folate receptor (RFC), and the folate receptor α (FR-α). In the cytoplasm, pemetrexed is polyglutamated by folylpolyglutamate synthetase (FPGS) and inhibits the enzymes involved in DNA and RNA replication, i.e., thymidylate synthase (TS), dihydrofolate reductase (DHFR), and glycinamide ribonucleotide formyltransferase (GARFT). Overexpression of TS, DHFR, and GARTF leads to chemotherapy resistance. The enzymatic activity of the γ-glutamyl hydrolase (GGH) leads to the hydrolysis of the glutamate tails in the lysosome. The depolyglutamated form is thereafter exported out of the cell by members of the ABC transporters (i.e., MRP).

**Figure 3 cancers-13-03211-f003:**
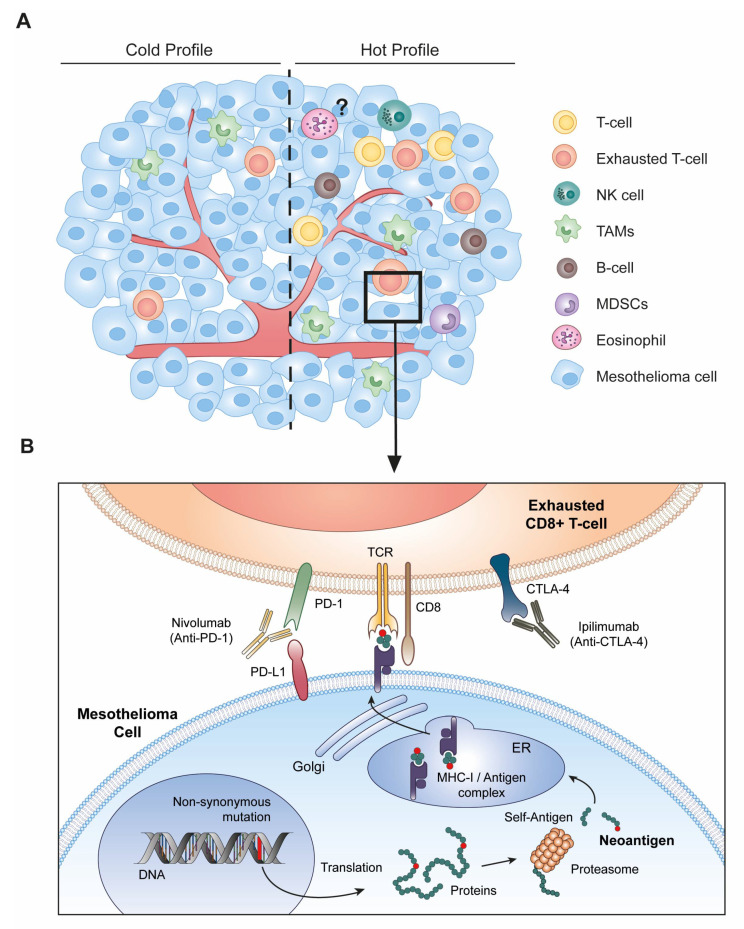
Immune checkpoint blockade therapy in the tumor microenvironment**.** (**A**) The heterogeneity of the malignant mesothelioma (MM) microenvironment results from the presence of endothelial, stromal, and immune cells (e.g., T-cells, tumor-associated macrophages (TAMs), and natural killer (NK) cells). MM is also characterized by spatial heterogeneity, displaying a continuum between “hot” and “cold” profiles (separated by the dotted line), defined by high and low lymphocyte infiltrations, respectively. (**B**) Non-synonymous mutations generate altered proteins, which are degraded by the proteasome. The resulting neoantigens are subsequently loaded on the major histocompatibility complex (MHC)-I in the endoplasmic reticulum (ER) and presented at the cell surface. These neoantigen-MHC-I complexes are recognized by the T-cell receptor (TCR) of primed CD8+ T-cells and elicit a cytotoxic response. However, T-cell function may be impaired after chronic stimulation of the TCR or by signaling of co-inhibitory factors. Exhausted CD8+ T-cells overexpress co-inhibitory receptors such as programmed-death 1 (PD-1) and cytotoxic T-lymphocyte 4 (CTLA-4). Treatment of MM patients with anti-CTLA4 (Ipilimumab) and/or anti-PD-1 (Nivolumab) antibodies tempers CD8+ exhaustion by blocking the co-inhibitory receptors thereby maintaining potent effector functions.

**Figure 4 cancers-13-03211-f004:**
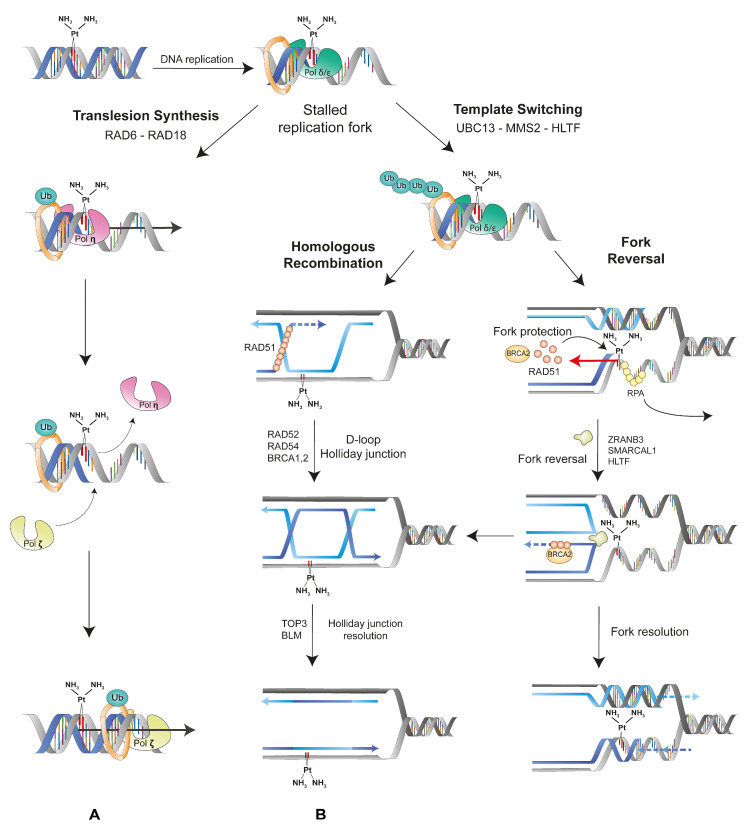
Activation of the DNA damage tolerance pathways by unrepaired cisplatin adducts. Unrepaired cisplatin adducts stall the replication fork during DNA synthesis and, if unrepaired, activate the DNA damage tolerance (DDT) pathways. (**A**) Mono-ubiquitination of proliferating cell nuclear antigen (PCNA) by RAD6-RAD18 leads to translesion synthesis (TLS). The Pol η translesion polymerase incorporates a nucleotide in front of the bulky adduct and bypasses the DNA lesion. Then, Pol ζ (REV3) further extends the distorted DNA strand. Poly-ubiquitination of PCNA by the UBC13-MMS2-HLTF complex leads to template switching and either (**B**) homologous recombination (HR) or (**C**) fork reversal (FR). In BRCA1/2-controlled HR, RAD51 promotes DNA strand invasion and RAD52 annealing. Then, RAD54 forms a D-loop and Holliday junctions that are resolved by helicase BLM and topoisomerase TOP3. Template switching may also proceed to FR upon processing by helicases ZRANB3, SMARCAL1, or HLTF after removal of the replication protein A (RPA) from single-strand DNA and its replacement by RAD51 and BRCA2. Resolution of the fork is finally processed either by HR or branch migration.

**Figure 5 cancers-13-03211-f005:**
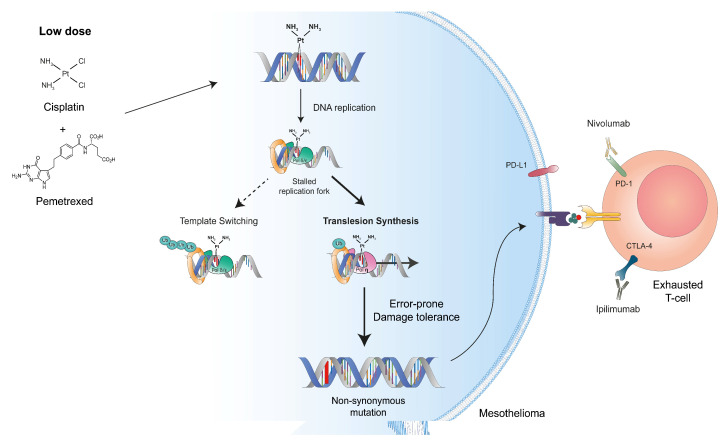
The concept of DDT induction combined with immune checkpoint inhibition. The paradigm is based on the administration of DNA-damaging drug to transiently activate the intrinsic mutagenic DDT pathway. Suboptimal doses of these compounds would activate error-prone DDT pathways (i.e., TLS) and increase random mutations. Providing that these mutations induce non-synonymous changes in coding genes, the newly formed peptides could be processed and presented by the MHC-I complex. Appearance of these neoantigens would occur in the absence of significant toxicity for immune cells.

## Data Availability

Publicly available datasets analyzed in this study can be found on the TCGA Research Network.

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
