# Peer review of "Activation of DNA Damage Tolerance Pathways May Improve Immunotherapy of Mesothelioma"

_cancers, 2021, doi:10.3390/cancers13133211_

Round 1

Reviewer 1 Report

this opinion paper presents a very interesting hypothesis to improve anti-tumor immunity, especially in cancers with a low mutation rate like mesothelioma.

However, I have not been able to find scientific evidence to support this hypothesis, neither in other tumors nor in other models, especially for cisplatin.

It is probably my fault and supporting evidence should be provided for the correct evaluation of your work.

Although it is an interesting hypothesis and certainly to be taken into consideration for subsequent research, the lack of data to support this hypothesis remains an extremely limiting factor.

Author Response

We agree that there is limited data available to support the hypothesis. This is why we proposed this manuscript as an "opinion" rather than a "review". However, evidence from the literature is becoming available. For example, transient activation of DDT coupled with immune checkpoint blockade is efficient in mouse melanoma (Zhuo et al, 2020). To further support the model, we have now included a series of papers showing that cisplatin induces TLS polymerase eta in non-small cell lung cancer (Ceppi et al, 2009). Another paper by Teng et al (2010) further shows that oxaliplatin induces TLS in gastric adenocarcinoma. Finally, the mutagenic activity of common cancer cytotoxic drugs has been compared (Szikriszt et al, 2016). Among these, cisplatin, carboplatin and oxaliplatin are mutagenic (Szikriszt et al, 2021). These references are now included in the new version of the manuscript.

Reviewer 2 Report

In this review article by Brossel et al., the authors discuss the opportunity to induce a transient activation of the error-prone DDT in order to generate more neoantigens in malignant mesothelioma. In line with this consideration, the authors suggest as novel therapeutic strategy in this context the combination of low doses chemotherapy with ICI immunotherapy. Although the purpose is very interesting, the present review does not sufficiently highlight the role of DDT pathway in anticancer immune response activation. The authors explained in details the molecular mechanism of DDT activation and function, but they did not mention papers in oncology that strongly support DDT role in neoantigen generation and immune anticancer responses. Therefore, I am not certain whether this review has achieved the goal they set up originally.

Finally, in some points I am not sure that the references cited are correct, for example ref.49 at line 408

Author Response

We agree that the manuscript does not sufficiently highlight the role of DDT pathway in anticancer immune response activation. Therefore, we proposed the manuscript to be published as an "opinion paper" rather than a review. As requested, we have now included a series of references that support the ability of low dose chemotherapy to induce error-prone TLS (Ceppi et al, 2009; Teng et al, 2010). We also added two reports on the mutagenic activity of cytotoxic drugs (Szikriszt et al, 2016 and 2021). Finally, we have better explained that transient activation of DDT coupled with immune checkpoint blockade is efficient in mouse melanoma (Zhuo et al, 2020).

It is correct that reference 49 is not adequate at line 408. We have moved this reference to the next sentence and included two reviews (Hato et al, 2014 and Mehmood 2014).

Reviewer 3 Report

The manuscript is well written and presented. Nicely prepared illustrations make it easy to understand and read.

The idea of combining low dose chemotherapy with immunotherapy is very interesting.  Authors should consider writing about the current DNA damage repair drugs (such as PARP inhibitors, see Triparna Sen et al 2019 in Cancer Discovery Journal, doi: 10.1158/2159-8290.CD-18-1020 ).

These would certainly have less toxicity issues, being synthtetic lethal for tumor cells.

Authors should also consider  the approach of targeting DNA repair proteins such as mth-1 (Gad et al , 2014) that are phenotypic lethal for tumor cells and mesothelioma.

Author Response

Thank you for these suggestions that will improve the manuscript. As requested, we have now included the PARP inhibitors (Triparna Sen et al 2019) and anti-mth-1 compounds (Wahi, 2021). This is now indicated in the manuscript on page 13.

Round 2

Reviewer 1 Report

Dear authors,

thank you for your reply.  In my opinion, the addition of these new references better supports your hypothesis adding value to the work.

Reviewer 2 Report

I agree with authors about the possibility to publish the manuscript as as an "opinion paper" rather than a review.